# Antioxidant Potential of Exosomes in Animal Nutrition

**DOI:** 10.3390/antiox13080964

**Published:** 2024-08-08

**Authors:** Hengyu Jin, Jianxin Liu, Diming Wang

**Affiliations:** Institute of Dairy Science, MoE Key Laboratory of Molecular Animal Nutrition, College of Animal Sciences, Zhejiang University, Hangzhou 310058, China; jhy.steve@zju.edu.cn (H.J.); liujx@zju.edu.cn (J.L.)

**Keywords:** exosome, antioxidants, oxidative stress, animal nutrition, milk exosome

## Abstract

This review delves into the advantages of exosomes as novel antioxidants in animal nutrition and their potential for regulating oxidative stress. Although traditional nutritional approaches promote oxidative stress defense systems in mammalian animals, several issues remain to be solved, such as low bioavailability, targeted tissue efficiency, and high-dose by-effect. As an important candidate offering regulation opportunities concerned with cellular communication, disease prevention, and physiology regulation in multiple biological systems, the potential of exosomes in mediating redox status in biological systems has not been well described. A previously reported relationship between redox system regulation and circulating exosomes suggested exosomes as a fundamental candidate for both a regulator and biomarker for a redox system. Herein, we review the effects of oxidative stress on exosomes in animals and the potential application of exosomes as antioxidants in animal nutrition. Then, we highlight the advantages of exosomes as redox regulators due to their higher bioavailability and physiological heterogeneity-targeted properties, providing a theoretical foundation and feed industry application. Therefore, exosomes have shown great potential as novel antioxidants in the field of animal nutrition. They can overcome the limitations of traditional antioxidants in terms of dosage and side effects, which will provide unprecedented opportunities in nutritional management and disease prevention, and may become a major breakthrough in the field of animal nutrition.

## 1. Introduction

Oxidative stress arises from an imbalance between pro-oxidants and antioxidants, leading to disruptions in redox signaling and control mechanisms within biological organisms [1]. Although oxidative stress is integral to normal physiological activities such as immune response, development, and cell proliferation, it also precipitates various pathological conditions, including cardiovascular diseases and insulin resistance, when oxidative damage results from an excessive accumulation of oxides [1,2,3,4,5]. In the area of livestock, excessive oxidative stress adversely affects animal health and performance, such as causing conditions such as mastitis in dairy cows, diminishing the quality of meat and dairy products [6]. Consequently, modulating oxidative stress through moderate antioxidant supplementation in animal diets is both effective and essential.

Exosomes, small vesicles released by cells via exocytosis, contain proteins, lipids, RNA, and other biomolecules and facilitate intercellular communication [7]. These nanoscale vesicles have been shown to play significant roles in processes such as immune response, disease treatment, and tissue repair [8]. Recent studies have elucidated the antioxidant properties of exosomes, demonstrating their ability to regulate intracellular redox balance and enhance the cellular antioxidant defense system by reducing reactive oxygen species (ROS) production, thereby exerting anti-inflammatory and cytoprotective effects [9]. Furthermore, exosomes can modulate cellular energy metabolism and oxidative stress responses by interpreting mitochondrial function [10]. These insights provide a deeper understanding of the biological functions of exosomes and their mechanisms of action in disease control.

Exosomes exhibit promising potential in the treatment of various diseases due to their diverse biological activities. Specifically, their antioxidant properties can be leveraged in feed processing to enhance animal performance and product quality by modulating redox responses and improving health status of the animals [11]. As novel antioxidants, exosomes play a crucial role in preventing and treating nutrition-related diseases in farm animals, such as fatty liver, ketosis, and negative energy balance, by mitigating oxidative stress damage induced by hazardous components in the feed. In summary, exosomes could serve as innovative antioxidants, bolstering disease resistance in animals and enhancing the quality of animal-derived products [12]. Given the superiority of exosomes in alleviating oxidative stress, future research will focus more on their nutrient delivery and corresponding metabolic regulation, further clarifying the mechanisms by which they regulate oxidative stress in the body. In addition, due to the significant role of exosomes, selecting which sources of exosomes can be widely obtained and better applied is also a key focus of future research. These contents will be mentioned later in the text.

## 2. Oxidative Stress and Animal Metabolism

Oxidative stress arises from the excessive accumulation of ROS, including superoxide anion, hydrogen peroxide, and hydroxyl radicals, indicating an imbalance between oxidative and antioxidant mechanisms within tissues and cells [13]. The ROS are essential components synthesized through normal cellular metabolic processes, such as mitochondrial respiration. ROS generation is dominated by external factors such as environmental pollution, disease, or physical exertion [14]. The over-accumulation of ROS leads to lipid peroxidation of cellular membranes and structurally damaged proteins and DNA, which can negatively impact cellular function and animal health [15]. Moreover, oxidative stress impairs the immune system of farm animals and diminishes their defending capacity against fetal microbes or viruses, which lead to death or reduce performance [16,17]. In addition, oxidative stress leads to dysfunction of nutrient absorption and utilization via impairing enzyme activity and intestinal mucosa integrity, thereby lowering their feed efficiency [18,19]. Therefore, understanding and managing oxidative stress is crucial for improving animal health and nutritional status, impacting both animal welfare and the efficiency and economics of the livestock system.

### 2.1. Role of Oxidative Stress in Animal Health Regulation

The role of oxidative stress on animal health manifests on multiple levels. Firstly, ROS impairs immune response by disrupting the membrane structure of several immune cells and related molecular signaling, thereby preventing secretion of cytokines. For example, stress diminished the proliferative rate of T cells and phagocytic activity of macrophages, which reduced the pathogen clearance efficiency of the tissue [20]. Furthermore, oxidative stress inhibits cellular growth rate and tissue development of mammalian animals via inducing cell death and inhibiting cell proliferation/differentiation [21]. The possible mechanism is that ROS accumulation activates various signaling pathways, such as tumor protein 53 (TP 53), nuclear-factor-activated B-cell kappa-light-chain enhancer of activated B cells (NF-κB), and the mitogen-activated protein kinase (MAPK) pathway, which are involved in cellular stress regulation, cell cycle control, and programmed cell death [22]. Consequently, persistent oxidative stress can reduce tissue regeneration and have long-term adverse effects on animal growth and development. Oxidative stress also plays a dominant role in animal reproduction regulation. Studies have shown that ovarian and testicular function can be disrupted via inhibiting hormone synthesis and secretion and, finally, reduce gamete quality and fertility [23]. For instance, over-accumulation of ROS in spermatozoa leads to serious lipid peroxidation, disrupts sperm membrane integrity and viability, and hinders their binding to oocytes consequently [24]. For female individuals, oxidative stress would cause poor embryonic development and various pregnancy complications [25].

In addition, oxidative stress interferes with nutrient metabolism through several pathways. Firstly, oxidative stress disrupts tissue integrity and functionality of the digestive tract. Excessive ROS produced by intestinal cells increase tissue permeability, nutrient extravasation, and facilitates invasions of hazardous substances [26]. Moreover, the high activities of digestive enzymes are essential for high-efficiency animal production, with enzyme dysfunction by ROS reducing the hydrolysis efficiency of macro-nutrients such as proteins, fats, and carbohydrates and consequently contributing to a lower total efficiency of feed mass [27]. Secondly, oxidative stress exacerbates the oxidative degradation and loss of antioxidant micronutrients, such as vitamins and minerals. The dietary micronutrients are the primary antioxidant source for mammalian animals and substances such as vitamins C and E are preferentially oxidized by ROS for antioxidant purposes. In addition, minerals such as zinc and selenium, as cofactors of antioxidant enzymes, also exhibit a decrease in levels under oxidative stress conditions [28]. In addition, oxidative stress can disrupt the balance and diversity of the gut microbiota. High doses of ROS can kill bacteria such as *Lactobacilli* and *Bifidobacteria*, reducing their levels; in contrast, it will promote the growth of harmful bacteria such as *E. coli* and the imbalance of this microbial population can lead to intestinal ecological imbalance [29]. Due to the crucial role of gut microbiota in the digestion and absorption of nutrients, their dysbiosis can impair nutrient absorption and biotransformation, such as leading to a decrease in the synthesis of short-chain fatty acids in the gut [30]. Lastly, oxidative stress directly attacks bioactive molecules like proteins and lipids, causing oxidative damage, structural changes, and cross-linking, thereby reducing their bioactivity and utilization and affecting their physiological functions [31].

Oxidative stress plays a crucial and multifaceted role in animal physiological functions, not only having adverse effects on the immune system, cell growth and development, and the reproductive system but also significantly interfering with nutritional metabolism. Essentially, it is a key factor leading to various physiological imbalances and functional disorders in animals, involving changes in cellular signaling pathways, damage to biomolecules, and ecosystem imbalances. The in-depth exploration in this field is of great significance for a comprehensive understanding of animal health maintenance and disease development, providing a theoretical basis for targeted prevention and treatment strategies. Future research should focus more on exploring its interaction with other physiological processes, as well as effectively regulating and intervening at the systemic level to optimize and safeguard animal health.

### 2.2. Antioxidant Property of the Nutrients

In organisms, in order to resist damage from oxidative stress, organisms rely on the protective effects of endogenous and exogenous antioxidants [31]. Key nutrients, such as vitamins and minerals, lead their antioxidant effects through a variety of mechanisms to maintain cellular integrity and function. For example, vitamin E, a fat-soluble antioxidant that functions primarily in cell membranes, is reported to reduce oxidative stress via protecting cell membranes from oxidative damage by scavenging lipid free radicals and interrupting the chain reaction of lipid peroxidation [32]. Similarly, vitamin C, as a water-soluble antioxidant, can scavenge free radicals directly or act as an electron donor to regenerate vitamin E and other antioxidant enzymes such as glutathione peroxidase (GSH-Px) [33]. In addition, β-carotene and other carotenoids are another important class of dietary antioxidants that protect against oxidative damage by physically quenching singlet-linear oxygen species and capturing free radicals to prevent oxidative damage [34]. Selenium is an essential micronutrient for GSH-Px, an enzyme that plays a central role in cellular antioxidant defense and protects cells from oxidative damage by catalyzing the reduction of hydrogen peroxide and lipid peroxides [35]. Zinc is an essential component of superoxide dismutase (SOD). This enzyme catalyzes the disproportionation reaction of superoxide radicals and reduces the production of free radicals. Zinc is also involved in maintaining the balance of the intracellular antioxidant system, which is essential for maintaining normal cellular immune function and cell division [36]. Copper is an essential component of copper–zinc superoxide dismutase (CuZnSOD), an enzyme that plays a key role in scavenging superoxide radicals. The antioxidant effect of copper is directly related to its active site in SOD, which plays an important role in protecting cells from oxidative stress [37]. As for iron, its role in oxidative regulation in the organism is complex. Although it is a cofactor of antioxidant enzymes and has an antioxidant effect, excessive accumulation of iron produces highly reactive hydroxyl radicals, which, in turn, exacerbate oxidative stress [38,39]. Therefore, balancing the circulating iron concentrations is crucial for maintaining animal metabolic homeostasis.

In addition to the aforementioned nutrients, polyphenolic compounds, such as flavonoids, anthocyanins, and phenolic acids, are also considered as potential antioxidants [40]. These compounds regulate oxidative processes through various mechanisms, such as direct scavenging of free radicals, chelation of metal ions, and activation of intracellular antioxidant pathways [41]. Moreover, interactions between different nutrients and antioxidants play a key role in maintaining the cellular redox balance. For instance, the synergistic interactions between vitamins C and E can enhance their antioxidant efficiency [42]. Thus, a comprehensive understanding of the antioxidant effects of key nutrients and antioxidants is essential for designing effective nutritional intervention strategies, enhancing antioxidant capacity of the tissue or cells, and maintaining their health status.

### 2.3. Advantage and Disadvantage of Traditional Antioxidant Approaches

Several nutritional strategies have been developed to regulate oxidative stress. Firstly, antioxidant supplementation is a direct means of alleviating oxidative stress. Adding antioxidants to feed can alleviate health issues caused by oxidative stress [43]. Small-molecule antioxidants such as vitamins C and E can directly eliminate free radicals in the host by donating electrons or hydrogen atoms, preventing cell damage [44]. Enzymatic antioxidants such as SOD and catalase catalyze the conversion of harmful ROS into harmless substances, enhancing the antioxidant defense ability of animals [45]. In addition, polyphenolic antioxidants such as catechins in green tea and resveratrol in grapes not only directly eliminate free radicals but also enhance the antioxidant capacity of animal cells by regulating cellular signaling pathways and gene expression [46]. The addition of these antioxidants can both enhance host immunity and reduce metabolic disease incidence, and it contributes to higher production efficiency and improved quality [43]. At the same time, maintaining the nutritional balance is vital for oxidative stress. Excessive levels of polyunsaturated fatty acids (PUFAs) in the body can readily induce oxidative stress [47]. Therefore, adjusting the ratio of PUFAs to saturated fatty acids (SFAs) in the diet can mitigate oxidative stress [48]. Additionally, meeting requirements of vitamins A, C, and E, as well as appropriate amounts of minerals such as copper, iron, and zinc, is essential for maintaining redox balance. These nutrients not only play a direct role in antioxidant defense but also indirectly influence oxidative stress by modulating immune responses and cell signaling pathways [49]. In addition, feed process protocol is critical for animal oxidative stress regulation. For instance, employing physical or chemical processes to protect the lipid components of feed, such as incorporating antioxidants or physical barriers to reduce lipid exposure to air, can minimize the formation of oxidation products [50]. Additionally, the inclusion of probiotics and prebiotics can enhance gut health, reduce endogenous ROS production, and improve gut barrier function by regulating the balance of the gut microbial community. This, in turn, can reduce inflammatory responses and oxidative damage [51].

In summary, oxidative stress can be effectively regulated through nutritional modulation strategies aimed at enhancing metabolic homeostasis [31]. The antioxidant effects of nutrients are influenced by an individual physiological state, nutritional status, and environmental factors [52]. Therefore, these strategies must consider individual variations and optimize nutrient doses and ratios. However, traditional antioxidant strategies face several limitations and challenges. Typically, traditional antioxidants are single molecular compounds with limited mechanisms for regulating oxidative stress, making it difficult to address the various physiological processes involved comprehensively [53]. Furthermore, antioxidants are usually administered orally or via injection, which complicates maintaining effective levels in the body over time, highlighting the need for improved bioavailability [54]. Notably, prolonged use of traditional antioxidant treatments can lead to tolerance and toxicity, posing health risks [55]. Future research should further investigate the effects of different nutritional strategies in specific environments and their interactions to enhance animal well-being. This includes precise control of nutrient dosages, in-depth study of feed additives, assurance of safety, and continuous optimization of oxidative stress management strategies in animal production systems. It is worth noting that, in recent years, the technology of utilizing exosomes secreted by cells has shown great potential [56]. These vesicles can transmit antioxidant molecules between cells, achieving more precise targeted regulation [57]. This exosome-based strategy not only significantly improves the effectiveness and safety of antioxidants, but also provides a promising solution for oxidative stress management, further promoting the overall well-being of animals [58]. Various approaches have been explored to cope with oxidative stress but traditional methods have obvious limitations. In the future, we need to focus on the depth, precision, and innovation, focus on the specific effects and interactions of different strategies, accurately control the nutritional dosage, innovate feed additives, ensure safety and efficiency, and optimize the overall management strategy. Particular attention should be paid to the potential of the emerging exosome technology in the field of antioxidants, which will open up new horizons for animal health management and promote the development of related research in a more comprehensive, precise, and cutting-edge direction, so as to realize the overall improvement of animal health and production performance.

## 3. Interplay between Exosomes and Oxidative Stress

### 3.1. Biogenesis, Release, and Transport of Exosomes

Exosomes are small nanoscale vesicles formed through the inward budding of the cell membrane and subsequent endocytosis [59]. Initially, early endosomes are produced through endocytosis and gradually form into late endosomes, a process that is mediated and influenced by the Golgi apparatus and endoplasmic reticulum [60]. These early endosomes mature into multi-vesicular bodies (MVBs), during which they invaginate to form small intraluminal vesicles (ILVs) containing cytoplasmic and organelle components [61]. The formation of ILVs is a complex process regulated by multiple molecular mechanisms, primarily through the Endosomal Sorting Complexes Required for Transport (ESCRT)-dependent and ESCRT-independent pathways [62]. In the ESCRT-dependent pathway, ESCRT-0 to ESCRT-III protein complexes recognize and aggregate specific cellular components to encapsulate into ILVs [63]. In contrast, the ESCRT-independent pathway involves liposome activity [64]. Upon fusion of the MVB with the cell membrane, ILVs are released into the extracellular microenvironment as exosomes. This release process is regulated by various factors, including Rab proteins, SNARE proteins, and calcium ions [65]. Once released, exosomes can be taken up by neighboring or distant cells through endocytosis, where they are internalized and processed [66] (Figure 1). Exosomes exhibit high stability and migratory capability within the humoral circulatory system, enabling their delivery over long distances via blood and lymphatic systems, and even allowing them to cross the blood–brain barrier [67]. This property renders exosomes highly effective carriers for biologically active molecules, offering significant potential for delivery applications. Exosomes are produced and released by almost all cell types, endowing them with diverse regulatory functions in both physiological and pathological processes [68]. The properties of widespread production and multi-functionality make exosomes a valuable biological resource and a focal point of recent research in the field of animal feed or human food.

### 3.2. Regulation of Exosomes by Oxidative Stress

#### 3.2.1. Effect of Oxidative Stress on Exosome Production

Oxidative stress plays a central regulatory role in exocytosis. For instance, low concentrations of hydrogen peroxide (H_2_O_2_, 5–100 μM) have been shown to promote exosome secretion in lens epithelial cells, human embryonic kidney 293 cells, and leukemia/lymphoma T and B cells [69,70,71]. Moreover, platinum nanoparticles also enhance intracellular ROS levels in human lung epithelial adenocarcinoma cells, thereby promoting exosome production and biomass release [72]. Furthermore, activation of the Ca^2+^–NAD (P) H oxidase 5 (NOX5) signaling axis by calcium ions accelerates exosome release and inhibits their uptake by vascular smooth muscle cells (VSMCs) [73]. Additionally, oxidative stress induced by 80 mM ethanol (EtOH) stimulates exosome release, although varying concentrations of EtOH differentially affect exosome levels containing proteins such as B-cell lymphoma 2 ssociated X protein, B-cell lymphoma 2 and autophagy related 12 [74,75]. Moreover, oxidative stress alters exosome formation by disrupting cell membranes and organelles, which are crucial for exosome packaging and transport [76]. Cellular membrane peroxidation by oxidative stress compromises membrane integrity and function. Oxidants like lipid peroxides inhibit cell proliferation by interfering with growth factor signaling at cell membrane sites, subsequently inhibiting DNA replication and mitosis and causing notable changes in cell morphology [77]. Moreover, oxidative stress can affect intercellular adhesion and polarity by modulating E-cadherin expression at the cell membrane, which is essential for epithelial cell morphogenesis [78]. Lysosomal function, regulated by cellular oxidative stress, is also linked to exosome production. For instance, excessive ROS production in podocytes following homocysteine (Hcy) stimulation inhibits lysosomal Ca^2+^ release through the transient receptor potential mucolipin 1 (TRPML1) channel. This inhibition prevents Ca^2+^-dependent lysosomal transport and lysosomal-MVB interactions, thereby enhancing exosome secretion [79,80]. Transcription factor binding to IGHM enhancer 3 (TFE3), as a transcription factor, enters the nucleus from the cytoplasmic solution in the presence of dephosphorylation, thereby stimulating lysosomal biogenesis. Typically, TFE3 is phosphorylated by mTOR in the cytosol [81]. However, under oxidative stress, mTOR is inactivated by ROS, which triggers the dephosphorylation of TFE3, which generates a large number of aberrant lysosomes, resulting in a functional imbalance of lysosomes, impaired degradation of MVB, and enhanced exosome release [82]. Additionally, oxidative stress may disrupt mitochondrial structure and function, indirectly affecting lipid metabolism and transport and consequently altering lipid content in exosomes [83]. Oxidative stress also modulates exosome production by regulating intracellular signaling pathways [84]. For example, nuclear factor erythroid 2-related factor 2, a transcriptional regulator activated by oxidative stress, translocates to the nucleus and activates the transcription of various genes involved in exosome assembly and release [85]. Furthermore, oxidative stress can indirectly impact exosome composition and function by altering the intracellular redox status [86].

#### 3.2.2. Effects of Oxidative Stress on Exosome Profiles

In addition to exosome production, exosome profiles can be greatly modulated by oxidative stress. For example, certain proteins and lipids within exosomes may be susceptible to oxidation or degradation during ROS accumulation, altering their functional properties [86]. Similarly, nonprotein constituents of exosomes like acids playing crucial roles in intercellular communication are different in different stress statuses [87]. Lipid peroxides formed by intracellular lipids are an important source of exosomes. Similarly, protein profiles of exosomes are modulated by stress-mediated protein processes, including synthesis, folding, or degradation, and, consequently, protein damage or aggregation [86]. This may change intercellular signaling and interactions in the cells [88]. Lerner et al. [82] found that exosomes secreted by stress-induced nonpigmented ciliary epithelium were able to be taken up by pretreated trabecular meshwork cells. These exosomes can enhance the antioxidant capacity of this cell through three pathways. The first pathway is that, after uptake of exosomes, more nuclear factor erythroid 2-related factor 2 (Nrf2) can be found in the cytoplasm and nucleus of the cell. The second pathway is that exosomes down-regulate p-GSK3β and β-catenin, two key proteins of the Wnt signaling pathway in the cell. In the last pathway, several antioxidant genes, including *Sod1, Sod2, Gpx1*, and *Hmox1*, were significantly up-regulated by the addition of exosomes, while the activities of CAT and SOD were enhanced in the cells. Regarding nonprotein components, oxidative stress may compromise the stability and function of nucleic acids by oxidative damage of glycopeptide bonds or bases. This oxidative damage may lead to nucleic acid degradation or dysfunction, affecting cell signaling and gene expression [89]. Carbohydrates are the key energetic sources for tissue metabolism; oxidative stress may disrupt intracellular carbohydrate metabolism pathways, resulting in changes in the concentration of carbohydrates such as glucose and fructose. These alterations reconstruct cellular energy supply and metabolic activities, thereby affecting cell viability and function [90].

#### 3.2.3. Effects of Oxidative Stress on Exosome Functionality

Oxidative stress modulates not only quantity and biomass composition of exosomes but also their functionality at a fundamental level. Specifically, oxidative stress can disrupt capacity of exosomes on immune response modulation and wound healing efficiency by altering specific molecular components within exosomes [84]. Additionally, oxidative stress enhances susceptibility of exosomes to degradation, thus diminishing their stability and efficacy within biological systems [83].

The general perception of ROS is as detrimental to biological systems in a dose-dependent manner. For example, low ROS concentrations play a beneficial role in wound healing by orchestrating cellular signaling pathways through redox mechanisms. For instance, ROS generated by cells associated with wound healing facilitate further repair of early acute wounds via signaling pathways involved in angiogenesis that are subject to redox regulation [91]. However, the dose-dependent regulation of ROS levels on exosomes and their intercellular communication differs substantially. In conditions of oxidative stress, exosomes may regulate immune responses and enhance wound healing by transporting specific signaling molecules, such as growth factors. These signaling molecules exhibit greater efficacy in environments with low ROS concentrations, as elevated ROS levels may impede their function or induce cellular damage [83]. Notably, while low ROS concentrations may promote wound healing, excessive ROS levels may impede tissue regeneration and remodeling, underscoring the importance of ROS concentration control in devising therapeutic strategies for oxidative stress management [92]. Furthermore, considering the role of exosomes in modulating immune responses, the judicious utilization of signaling molecules carried by exosomes may offer novel strategies for optimizing wound healing under oxidative stress conditions [93].

### 3.3. Role of Exosomes in Oxidative Stress Regulation

#### 3.3.1. Antioxidant Role of Exosomes Property

The involvement of exosomes in cellular responses to oxidative stress is intricate and multifaceted, as they intricately regulate oxidative stress by transporting crucial molecules such as message RNA (mRNA), proteins, microRNA (miRNA), and circular RNAs (circRNA) [94]. These exosomes are discharged by various cell types, encompassing both stem and differentiated cells, and their distribution and biological functions in vivo indicate a pivotal role in intercellular communication [95]. The mRNA molecules harbored within exosomes possess the capability to be translated into proteins with antioxidant properties upon uptake by recipient cells [96]. Notably, these proteins, such as SOD and GSH-Px, directly neutralize ROS, thereby reducing oxidative stress. This mechanism facilitates a prompt response to elevated redox status by bolstering cellular defenses through heightened expression of endogenous antioxidant enzymes [97]. Furthermore, exosomes convey proteins that encompass additional antioxidant molecules capable of directly counteracting ROS and mitigating their detrimental effects on cellular components [76]. The delivery of such proteins assists recipient cells in establishing or reinstating redox homeostasis, particularly in oxidative stress status [98]. For example, heat shock proteins (HSPs) in exosomes play an important role in coping with oxidative stress. The expression level of HSPs increases when cells are exposed to heat stress and oxidative stress, which can protect proteins in cells from damage and help repair damaged protein structures. These protected proteins can, in turn, enhance the antioxidant capacity of cells and reduce cell apoptosis and inflammatory responses caused by oxidative stress [99].

miRNAs play a pivotal role in modulating oxidative stress. A previous study suggested that encapsulated miRNAs coated in exosomes can ameliorate oxidative stress by modulating the expression of specific genes [100] (Figure 2). For instance, miRNAs may down-regulate the expression of enzymes from the NOX family, which serve as pivotal sources of ROS generation [101]. miR-320a in exosomes of human amniotic MSCs can down-regulate the expression of sirtuin 4 (SIRT4) by targeting the untranslated region of SIRT4 mRNA and thus reduce the generation of ROS [102]. miRNAs can also alleviate oxidative stress in the body by regulating intracellular ion homeostasis. A typical example is miR-23a-3p and miR10a-5p in exosomes, which can reduce ROS levels by inhibiting the expression of divalent metal transporter 1 (DMT1) and cartilage acid protein 1 (CRTAC1), respectively [103,104]. Additionally, miRNAs may fortify cellular antioxidant defense mechanisms by activating protective signaling pathways, such as NF-κB, phosphatidylinositol-3-kinase/protein kinase B (PI3K/Akt) signaling pathway and Nrf2/antioxidant response element (ARE) [105]. circRNAs, as a novel class of noncoding RNA, contribute to oxidative stress regulation via interacting with other RNA molecules [106].

Research has shown that circRNAs in exosomes regulate oxidative stress through various mechanisms. For example, circHIPK3 is significantly down-regulated under oxidative stress conditions induced by hydrogen peroxide (H_2_O_2_), but its overexpression can alleviate cell death caused by H_2_O_2_, indicating that circHIPK3 has a protective effect in antioxidant stress [107]. In addition, circZNF609 regulates the c-Jun N-terminal kinase and p38 mitogen-activated protein kinase signaling pathways by adsorbing miR-145 to prevent its binding to target molecules, thereby protecting cells from oxidative stress induced by H_2_O_2_ [107].

These diverse functions of exosomes not only advance our understanding of intercellular communication in oxidative stress regulation but also present promising targets for the development of novel therapies against oxidative-stress-related diseases. Consequently, exosomes assume a multifaceted role in the modulation of oxidative stress, emphasizing the importance of investigating their role in maintaining cellular redox balance and production enhancement. Future research endeavors should further elucidate the specific mechanisms underlying exosome-mediated actions in various disease states and explore their potential utility as therapeutic modalities to intervene in oxidative-stress-related pathological processes.

#### 3.3.2. Targeted Antioxidant Efficiency of Exosomes

In recent years, exosomes have enormous potential as carriers for delivering antioxidant molecules for nutritional strategy. The inherent bilayer membrane structure of exosomes affords them high biocompatibility and stability and protects encapsulated antioxidant molecules from being degraded and activated [108]. Furthermore, the nanoscale size of exosomes facilitates their traversal of biological barriers, enabling targeted delivery [109]. Exosomes are readily obtainable from various cell lines and body fluids, providing a foundation for their large-scale production [110]. Yao et al. [111] observed that exosomes derived from human umbilical cord mesenchymal stem cells (huc-MSCs) attenuated carbon-tetrachloride-induced liver injury when administered via tail vein injection. These exosomes contained GSH-Px, a key enzyme in regulating oxidative stress and apoptosis, which is proved by GSH-Px knockdown experiment. Cui et al. [112] developed a targeted delivery system that enhanced the cellular adaptability of astaxanthin by modifying exosomes with hyaluronic acid, improving cellular uptake and stability in a macrophage model, and effectively suppressing inflammatory factors. Similarly, Tian et al. [113] engineered exosomes by attaching the cyclic Arginine-Glycine-Aspartic acid-Tyrosine-Lysine peptide for targeted delivery to ischemic brain regions, which can inhibit inflammation and apoptosis with loaded curcumin in an in vivo cerebral ischemia model. Kim et al. [114] provided an overview of exosomes in drug delivery systems and highlighted non-immunogenicity, high biocompatibility, and exceptional drug-carrying capacity of the exosomes. Song et al. [115] summarized advances in using exosomes as carriers for natural product delivery, highlighting their role in enhancing the targeted delivery and therapeutic effects of compounds like paclitaxel and curcumin. These studies underscore the potential and efficacy of exosomes as delivery vehicles for antioxidants and other bioactive compounds. Through engineering and surface modification, exosomes facilitate efficient and targeted drug delivery, offering new strategies for treating various diseases.

## 4. Antioxidant Potential and Application of Exosomes in Animal Nutrition

In the context of animal nutrition, the antioxidant potential of exosomes is primarily demonstrated in three ways. Firstly, exosomes can carry antioxidant enzymes such as SOD and GSH-Px, which are integral to intracellular antioxidant reactions, scavenging ROS and protecting cells from oxidative damage [116]. miRNAs and mRNAs contained within exosomes can modulate gene expression in recipient cells, thereby enhancing their antioxidant capacity [9]. Moreover, exosomes serve as carriers of trophic factors, including growth factors and cytokines, which promote the repair and regeneration of damaged tissues [117]. Secondly, exosomes are a promising application in animal nutrition as delivery systems for drugs or nutrients. Their biocompatibility, stability, and targeting properties make exosomes effective vehicles for improving the bioavailability and efficacy of therapeutic agents or nutrients [118]. Thirdly, antioxidant effects of exosomes in animal nutrition are further reflected in their ability to regulate and enhance the body’s immune response, particularly concerning anti-inflammatory and antioxidant activities [119].

### 4.1. Advantages of Exosomes as Antioxidants in the Field of Animal Nutrition

Considering the limitations of traditional nutritional antioxidant strategies against oxidative stress, it is promising to develop exosomes as novel antioxidants in animal nutrition. With their unique physiological characteristics and functional properties, exosomes offer enhanced antioxidant capacity that addresses the shortcomings of conventional antioxidants [120]. Due to their single-component mechanisms, conventional antioxidants are often insufficient to comprehensively mitigate complex oxidative stress damage. In contrast, exosomes are multifunctional, containing a variety of active components capable of effectively addressing oxidative stress [121]. For instance, exosomes derived from mesenchymal stem cells have been shown to reduce oxidative damage by regulating apoptosis and promoting cell regeneration through various miRNAs [122]. In addition, prolonged or high-dose use of traditional antioxidants can lead to adverse side effects [123]. For example, excessive vitamin E intake is associated with an increased risk of bleeding, and high β-carotene levels are positively correlated with lung cancer risk rate of smokers [124,125]. Due to their superior biocompatibility, exosomes minimize the risk of immune rejection [118]. Traditional antioxidants have a lack of targeted delivery; modified exosomes can be precisely directed to target tissues or cells, enhancing their therapeutic efficacy [126]. For example, exosomes carrying cardioprotective factors have significantly reduced cardiomyocyte apoptosis and improved myocardial function [127]. Furthermore, traditional antioxidants are quickly metabolized in various tissues and fail to maintain effective levels over extended periods [128]. Owing to their nanostructures, exosomes can maintain the targeted biomass level in the targeted cells for a long term [129]. This sustained-release approach overcomes the rapid clearance issues associated with conventional antioxidants.

### 4.2. Physiological Heterogeneity of the Antioxidant Capacity of Exosomes

The composition and origin of exosomes are dynamic, and their physiological heterogeneity at the antioxidant level is decided by the donor cells from which they originate and their intrinsic composition [130]. The heterogeneity allows exosomes to perform various roles in different physiological states, adapting to complex scenarios [131]. Exosomes derived from neural stem cells are rich in miRNAs that are integral to molecular mechanisms promoting cell regeneration, making them effective in repairing oxidative damage in the nervous system [132]. Similarly, exosomes derived from mesenchymal stem cells (MSCs), particularly bone marrow MSCs, exhibit significant antioxidant capacities [133]. These exosomes can markedly reduce ROS accumulation, and inhibit inflammatory factor expression [134]. Moreover, bone marrow MSC-derived exosomes can alleviate oxidative damage to neurons and enhance survival rate of nerve cells [135]. Exosomes from vascular endothelial cells have been shown to significantly mitigate endothelial damage, enhance SOD activity, and reduce myeloperoxidase (MPO) expression in models of lipopolysaccharide-induced endothelial cell dysfunction, thereby shielding endothelial cells from oxidative stress [136]. Conversely, exosomes derived from tumor cells, which contain pro-oncogenic factors such as transforming growth factor-βand hypoxia-inducible factor-1α, can exacerbate cancer cell invasion and metastasis, thus deepening oxidative damage [137]. Therefore, to fully harness the therapeutic potential of exosomes in combating oxidative stress, it is crucial to thoroughly understand their origin, intrinsic components, and physiological heterogeneity.

### 4.3. Antioxidant Applications of Advantageous Exosomes in Animal Nutrition

Exosomes from various sources are important in oxidative stress regulation. It has been revealed that exosomes derived from fruits, including blueberries, strawberries, and pomegranates, can effectively mitigate oxidative damage through diverse regulatory mechanisms [138,139,140]. In the context of animal-derived exosomes, these vesicles can be extracted from tissues and body fluids, which can be a route to enhance animal health in a natural, green, healthy, and sustainable manner, such as milk exosomes [141] (Figure 3). Milk-derived exosomes are readily available, cost-effective, and can be produced on a large scale [142]. Their biocompatibility and low immunogenicity reduce the risk of immune reactions, while their lipid bilayer structure provides natural protection for active ingredients, enhancing stability and bioavailability [143]. Notably, milk exosomes contain resistant glycoproteins (xanthine dehydrogenase, butyrophilin and mucin 1) and surface proteins (flotillin 1, intercellular adhesion molecule 1, apoptosis-linked gene 2 interacting protein X and epithelial cell adhesion molecule), which confer resistance to pepsin and tolerance to acidic gastric environments [144]. This ensures their integrity and efficient entry into the bloodstream via endocytosis in the gastrointestinal tract, paving the way for new oral drug delivery methods [145]. Additionally, milk exosomes have the potential to cross biological barriers, facilitating drug delivery to traditionally challenging regions such as the intestinal epithelial and vascular endothelial barriers [146]. These properties make milk exosomes promising candidates for disease treatment and prevention, particularly in enhancing drug efficacy and reducing side effects. In terms of immunomodulation, exosomes extracted from commercial milk have demonstrated the ability to inhibit lipopolysaccharide (LPS)-induced inflammatory responses [147]. Bovine milk of various sources and treatments assessed their impact on inflammatory factors released by macrophages [148].

In addition, milk-derived exosomes offer several advantages over traditional antioxidants. First, milk contains a broad spectrum of both water- and fat-soluble antioxidants, as well as specific enzymes and proteins such as lactoferrin and sulfur-containing nitrogen compounds, which are less common in other food sources [149]. The antioxidant system in milk is naturally occurring, providing a complex and synergistic composition that surpasses the effectiveness of artificially added antioxidants [150]. While conventional antioxidants such as vitamin E and β-carotene are effective at neutralizing free radicals, they often require dietary supplementation and may exhibit limited absorption and utilization efficiency in animals [151]. In contrast, milk exosomes are natural bioactive substances, which can be directly absorbed and utilized by animals, thereby enhancing the bioavailability of antioxidants. Moreover, milk exosomes are rich in fatty acids, proteins, and other immunity boost molecules that enhance their antioxidant properties [142] (Figure 3).

## 5. Conclusions

In summary, the incorporation of exosomes into animal nutrition provides a more efficient source of antioxidants and delivers comprehensive nutritional support and health protection through their diverse bioactive components, suggesting exosomes as a novel nutritional supplement with extensive applications in animal nutrition and health management. Focusing on industrial issues, delving deeper into how exosomes leverage their antioxidant properties to slow down food oxidation and minimize its detrimental effects on food nutrition and texture, emerges as a pivotal strategy in future food preservation research. However, to ensure large-scale production, quality control, and safety assessment of exosomes, further development of appropriate standards is necessary and extensive research must continue in this field.

## Figures and Tables

**Figure 1 antioxidants-13-00964-f001:**
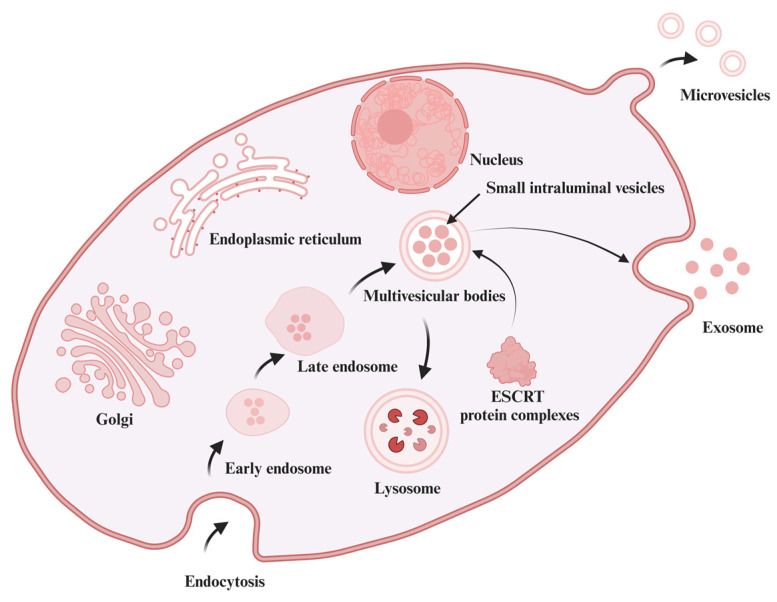
The mechanism of exosome production. Cells produce vesicles through endocytosis, which fuse to form early endosomes and gradually become late endosomes, accompanied by the mediation of Golgi apparatus, endoplasmic reticulum, and nucleus. Subsequently, the late endosomes produce many luminal vesicles (ILVs) in the cytoplasm, which gradually evolve into multi-vesicle bodies (MVBs). Finally, it is often accompanied by Endosomal Sorting Complexes Required for Transport (ESCRT) protein complexes, which are released outside the cell to form exosomes.

**Figure 2 antioxidants-13-00964-f002:**
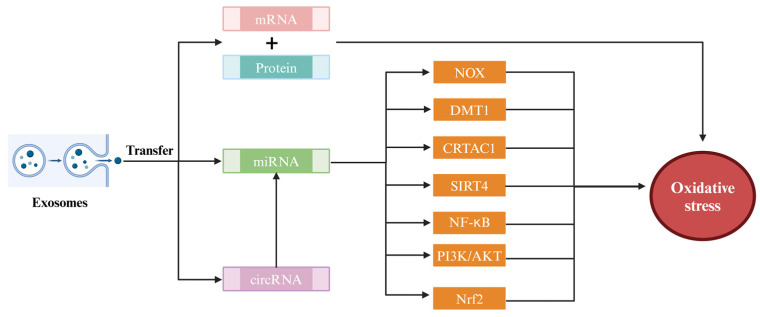
Multiple pathways through which internal components of exosomes alleviate oxidative stress.

**Figure 3 antioxidants-13-00964-f003:**
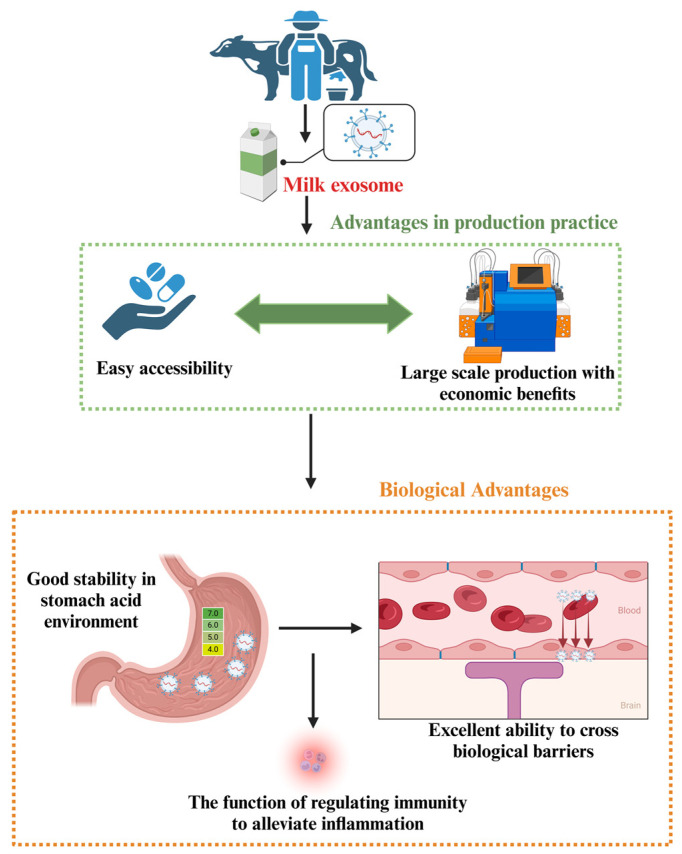
The potential of milk exosomes in the field of animal nutrition.

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
