# Peer review of "Antioxidant Potential of Exosomes in Animal Nutrition"

_antioxidants, 2024, doi:10.3390/antiox13080964_

Round 1
Reviewer 1 Report
REVIEW for the journal Antioxidants (ISSN 2076-3921)
Review “Antioxidant potential of exosomes in animal nutrition”
Manuscript ID: antioxidants-3104030
Authors: Hengyu Jin, Jianxin Liu, Diming Wang
Brief summary. In my opinion, this review is crucial for advancing both scientific knowledge and practical applications in animal nutrition. By exploring the potential of exosomes as novel antioxidants, it offers a promising avenue for overcoming the limitations of traditional antioxidants, thereby enhancing the efficiency and effectiveness of nutritional interventions in the feed industry.
Comments
1. Abstract.
I suggest that the authors provide a summarizing sentence at the end to highlight the significance of their review, especially since the abstract is not too long (157 words).
2. Introduction.
Here is a suggestion to improve the introduction:
Elaborate more on future research directions: Provide more detail on what specific aspects of exosomes should be studied further.
3. The structure of the review
is well-organized and logically progresses through key topics relevant to the subject. The sections are comprehensive and cover essential aspects necessary for a thorough understanding of the topic. The structure of the review includes the following sections: Oxidative Stress and Animal Metabolism. Interplay Between Exosomes and Oxidative Stress. Antioxidant Potential and Application of Exosomes in Animal Nutrition.
Proposal: Interplay Between Exosomes and Oxidative Stress: While this section covers the interactions between exosomes and oxidative stress, it could benefit from a more detailed explanation of the underlying biological mechanisms and pathways involved.
4. The conclusion
highlights the potential of exosomes as antioxidants for animal nutrition, but it lacks specific evidence and may overgeneralize their applications. Additionally, there is insufficient exploration of the biological mechanisms, production challenges, and economic aspects. The future research directions are also somewhat vague and could benefit from more detailed suggestions.
5. The reference list
should be carefully reviewed to ensure it complies with the journal’s specific requirements.
In my opinion, the work done by the authors is both relevant and innovative from a theoretical and practical perspective. The article is well-structured and offers a comprehensive review of the advantages of exosomes as novel antioxidants in animal nutrition and their potential for regulating oxidative stress. However, the review could benefit from minor adjustments to provide more specific evidence, address challenges for large-scale production, and offer more detailed future research directions.
Sincerely, Reviewer.
1. Line 20. A summary sentence is needed. 2. Lines 230-235. Could you specify the authors of Figure 1? 3. This question also applies to Figures 2 ( Line 421) and 3 (Line 478 ).Author Response
In response to the reviewer's comments, the uploaded file has been replied to in the relevant format. Please refer to the uploaded file for details.

Reviewer 2 Report
The topic of this literature review is of significant scientific interest to specialists in the field of medicine and veterinary medicine. The authors analyze the basis of the physiological effects of reactive oxygen species on various components of a living cell and physiological processes in the body as a whole.
Key notes:
1. The review is a collection of material written by different authors: some subsections are quite professional and correct, the wording does not allow the meaning to be misunderstood. Other subsections contain gross inaccuracies. The article requires very careful reading by all authors to eliminate numerous errors.
2. A good literature review allows readers to look at a problem from an unexpected angle, systematizes scattered publications, and directs further research on this topic in the right direction. The article under review is, on the one hand, a mixture of trivial facts and fragments from university textbooks on biochemistry and reference books on dairy production and, on the other hand, a set of truly new facts in the studied area. I recommend that authors reduce the volume of well-known facts in the first 60% of the article and deepen the analysis of specialized literature that clearly relates to the research topic.
3. I did not see deep structuring of the material in the form of diagrams and tables. Figure 1 can probably be left alone. Figure 2 should be removed because it does not show a sufficiently scientific comparison of objects. Figure 3 is more suitable for a textbook for introductory university courses. It's better to remove it. In general, this article needs to be expanded with diagrams of metabolic regulation processes, supplementing the text with tables and figures, which will be the result of the textual presentation of the material. In the process of preparing these figures, the authors themselves will see the incompleteness of the analysis in the text of some processes. The purpose of adding diagrams is not to write a chapter for a new student on biochemistry or physiology of humans and animals (textbooks show well-known facts), but to create diagrams that will help biochemists with many years of research experience be surprised to discover previously unstudied relationships.
Technical shortcomings of the manuscript.
1. Lines 57-71 and 383-395 should receive their own subsection title or should be included in a subsequent subsection.
2. Line 80, 81 and others: the first mention of even well-known proteins, genes and metabolic pathways in the article should be accompanied by their full name, and not just an abbreviation.
3. Line 102: a dubious statement that should be formulated more correctly and should be supported by reference to the literature. It is likely that, for example, the zinc atom or ion does not suffer in any way from the effects of ROS.
4. Line 107: the statement is doubtful. The most numerous intestinal bacteria are called “harmful bacteria” by the authors. This is a controversial statement, to say the least.
5. Line 110: What is “gut region”?
6. Line 111: why do the authors classify lipids as macromolecules?
7. The link to the source should be located immediately after the author's last name, and not at the end of the sentence (for example, lines 367, 370, 373, 375 and many others).
8. Before references to literature, a space is required everywhere (for example, lines 414, 416, 417).
9. Errors in punctuation (for example, lines 419, 448).
10. The literature is formatted very carelessly, not according to the journal standard (for example, line 805, 807). The names of the journal are not abbreviated according to the standard (the authors tried to put the first 30 literature sources in order, but even in these 20% of the bibliography there are a large number of errors - periods, spaces, commas, capital and non-capital letters). First and last names are not abbreviated according to the rules (for example, lines 775, 787). The initials are placed before or after the last name (for example, lines 594, 653, 697). In magazine titles, words are not capitalized (for example, lines 753, 754, 804, 809). The year, volume and number of the journal are not printed according to the rules (for example, line 759). DOIs were printed with errors (for example, line 773). In some cases, all the words in the title of the article are capitalized for some reason (for example, line 778). In some cases, out of many authors of an article, one author and others are indicated, in others - two authors and others, sometimes more than 10 authors are indicated. For some reason, the authors use quotes (line 627).
Author Response
In response to the reviewer's comments, the uploaded file has been replied to in the relevant format. Please refer to the uploaded file for details.

Reviewer 3 Report
The manuscript includes a very interesting review focused on the nature, interest and application of exosomes. According to the novelty of item and the way it has been treated, I think it can be accepted after some minor aspects are performed.
Abstract
Lines 9-11: Perform the sentence.
Clarify in this section if the review is focused on the oxidative stress in animals or in animal nutrition. Or on both. In my opinion, this fact is not clearly expressed in this section.
Keywords
Include extracellular vesicles.
1.Introduction
Lines 53-55: This sentence belongs to Concluiosns or Final Remarks section.
4. Antioxidant potential and applications
According to their high content on PUFAs and susceptibility to oxidation, marine species may require a special attention on exosomes presence and effects. Could the authors include any marine exosome examples ?
5. Conclusions
On-coming research on the employment of exosomes on food protection may be mentioned.
Author Response

(The authors gave the same response as above.)

Round 2
Reviewer 2 Report
The article has improved significantly. It is likely that once numerous technical deficiencies in the text and literature have been resolved, this article can be published.
The text needs to correct numerous unsuccessful formulations, lack of references to literature, and the numbering of these references. There are a lot of errors in the design of literature.
Author Response

(The authors gave the same response as above.)
